# Human activity recognition using magnetic induction-based motion signals and deep recurrent neural networks

Negar Golestani[1✉] & Mahta Moghaddam [1]

Recognizing human physical activities using wireless sensor networks has attracted significant research interest due to its broad range of applications, such as healthcare, rehabilitation, athletics, and senior monitoring. There are critical challenges inherent in designing a sensor-based activity recognition system operating in and around a lossy medium such as the human body to gain a trade-off among power consumption, cost, computational complexity, and accuracy. We introduce an innovative wireless system based on magnetic induction for human activity recognition to tackle these challenges and constraints. The magnetic induction system is integrated with machine learning techniques to detect a wide range of human motions. This approach is successfully evaluated using synthesized datasets, laboratory measurements, and deep recurrent neural networks.

---

[1] Ming Hsieh Department of Electrical and Computer Engineering, University of Southern California, Los Angeles, CA 90089, USA. ✉email: golestani.
negar@gmail.com

Human activity recognition (HAR) aims to provide information on human physical activity and to detect simple or complex actions in a real-world setting. It allows computer systems to assist users with their tasks and to improve the quality of life in areas such as senior care, rehabilitation, daily life-logging, personal fitness, and assistance for people with cognitive disorders[1–6]. Two main approaches for deployment of HAR systems are external and wearable sensors[7]. In the external approach, the monitoring devices are set at fixed points, and users are expected to interact with them[8]. The vision-based technique, for example, is one of the well-known external methods that has been extensively studied for human activity analysis[9,10]. However, it faces many challenges in terms of coverage, accuracy, privacy, and cost. It requires infrastructure support, such as the installation of video cameras in surveillance areas, which is usually costly. Additionally, cameras cannot capture any data if the user performs out of their reach[11,12]. In the second approach, on-body sensors, such as accelerometers, gyroscopes, and magnetometers, are used to translate human motion into signal patterns for activity recognition[13–15]. Recent advances in embedded sensor technology have made it feasible to monitor the user's activity using smart devices. Several research studies have reported the use of smartwatches and smartphones in human activity monitoring, and have presented a satisfactory performance[16–19]. Although these devices provide a privacy-aware alternative solution that overcomes many disadvantages of the external approach, they still might not be able to address the requirements of a diverse range of applications. A single wearable cannot cover the entire body and therefore fails to obtain adequate information about the mobility of all body segments[20–22]. For example, inertial sensors embedded in a smartwatch cannot capture the movement of legs, which restricts the capability of the system in classifying activities. Additionally, in systems relying on data from a single device, variations in position can have a significant effect on the performance or lead to the failure of the monitoring system[20,23,24].

Wireless body area network (WBAN) consisting of wearable devices operating around the human body can tackle these problems[21,25]. In WBANs, sensors are spatially distributed over the human body and collect data from the user. Then data are transmitted wirelessly to a central processing unit for detection. This approach can provide comprehensive information on the mobility of body segments and potentially improve system accuracy. However, WBAN design is challenging as many constraining, and often conflicting, requirements have to be taken into account[26–28]. For example, the system has to be inexpensive, accessible to the general public, and meet ergonomic constraints and health requirements. It has to operate under proper guidelines limiting the power exposure to the user since the energy absorption may lead to temperature elevation in biological tissues. To ensure users' safety, it has to satisfy specific absorption ratio (SAR) constraints, while providing a reliable wireless link[29]. Moreover, the system should guarantee the security and privacy of the user's data. Wearable devices must be small and lightweight, which puts a restriction on the battery size and longevity. On the other hand, frequent battery recharging may not be practical for sensor networks with multiple sensors in applications such as senior monitoring[7]. Due to the limitation of energy resources, the power management has become a critical issue in designing a WBAN. Since wireless communication consumes a considerable portion of the energy[30], numerous studies have proposed and investigated low-power solutions[31–34]. The conventional state-of-the-art wireless sensor networks working in the vicinity of the human body adopt radio-wave propagation for signal transmission. This technique is susceptible to the characteristics of the environment, and its signal experiences a high attenuation around a lossy medium, such as the human body. It results in higher power consumption, shorter battery life, and lower reliability[33,35,36]. Moreover, radio-wave propagation technologies are prone to interference with adjacent communication links since most of them, such as Bluetooth, operate at the busy 2.4 GHz, the industrial, scientific, and medical (ISM) band[37,38]. They also have potential security problems as their signal cannot be stopped from propagating into free-space. Therefore it can be intercepted even distant from the transmitter[39].

We introduce the magnetic induction-based HAR (MI-HAR) system that effectively detects physical movements by magnetic induction (MI) signals. This system represents the motion of human body parts via variations in the MI signals transmitted from transmitter to the receiver during physical action, instead of spatial data measured by the inertial sensors. This approach can overcome several problems associated with conventional sensor-based HAR systems, such as eliminating the need for an extra wireless module, reducing power consumption, and the required bandwidth by combining data collection and wireless signal transmission steps. Moreover, it has other features that are inherited from the MI-based communication system. Here we verify the capability of the proposed method in identifying human actions. We first synthesize MI motion data corresponding to several physical activities. Then we apply machine learning-based classifiers and deep recurrent neural networks to classify human movements. The results indicate that the MI signals are informative descriptors for the motion of human body parts.

## Results

**System principle**. The MI-based communication system is a short-range wireless physical layer that transmits signals by coupling non-propagating magnetic field between the wire coils rather than radiating as conventional methods. The main component of each node is a coil, which is lightweight, portable, inexpensive, simple, and can be worn as accessories such as belts, wristbands, and jewelry[33,40]. The manufacturing cost of an MI module is approximately less than \$20, while a Bluetooth IMU costs more than \$100 (refs. [41–43]). The MI coils have a small radiation resistance, which means that the energy propagated to the far-field is negligible. As a result, multipath fading is not an issue, and the MI system can offer a much better quality of service (QoS) compared to Bluetooth-type systems[33,44,45]. The non-propagating magnetic field produced by the coils falls off proportional to $r^{-3}$ instead of $r^{-1}$ for radiating fields at a transmission distance $r$. Although the rapid decay limits the coverage range, it can be favorable in short-range applications such as WBANs[46]. It allows the signal to remain in a 'bubble' around the coil, which provides a personalized space for the user. It also minimizes the leakage outside the targeted coverage range, reduces interference, increases security, and enables bandwidth reuse[44,47]. One of the main notable advantages of the MI system is that it works well in lossy dielectric media, such as the human body[48]. In these environments, the MI system experiences much less energy absorption compared to conventional radio-wave propagation technologies[49]. It results in lower SAR for applications working around the human body. Due to smaller path loss, the MI system can transmit a signal with much less power for the same range. This system can be up to six times more efficient in terms of battery power compared to other short-range communication systems (e.g., Bluetooth)[47]. This characteristic enables a large variety of novel and demanding applications in harsh environments such as underwater monitoring of scuba divers[39,49,50].

The signal generated by an MI coil attenuates as a function of frequency, channel medium, coils' geometry, location, and alignment (see Methods section)[33]. The non-propagating

magnetic field is mainly affected by the permeability of the medium, which is close to the air for non-ferrous materials. The MI channel condition remains constant even in an inhomogeneous lossy medium, such as around the human body[33,49]. For the frequency of up to 30 MHz, the dimension of the human body is relatively small compared to the wavelength, which makes the propagation and scattering effects insignificant[33]. The immunity of signal in this frequency range to the environment makes the forward voltage gain, $S_{21}$, of the MI system only a function of coils' locations and alignments for a predefined coil geometry and operating frequency. The gain varies by changing the distance and alignment between the MI coils, and therefore, relative motion between the MI coils yields patterns in the received MI signal. This unique characteristic of the MI system is the fundamental principle of the proposed MI-HAR system.

**System framework**. The activity recognition process steps are different depending on the application. The framework used in this paper has two main stages: data acquisition and detection. For the first stage, an MI-based communication system is employed, which enables the integration of sensing and wireless data transfer into a single step. The user wears the receiver (RX) coil, for example, as a belt around the waist, and transmitter (TX) coils can be placed around the other skeleton bones, such as wrists, arms, and legs. The human body bones are spatially translated and oriented during a physical activity, which changes the relative location and alignment of the MI coils around them. Collecting the received MI signals transmitted from the coils enclosing skeleton bones can model the relative motion of human bones to represent motion. Since the spatial variations of skeleton bones over time are discriminative descriptors of human actions[51], the vector of samples observed by the MI coils over time can be considered as the set of inputs for the activity detection algorithm. Increasing the number of coils around the skeleton bones results in a broader set of input data. It consequently enhances the accuracy of the MI-HAR system in detecting the relative motion of body parts. In the next step, a classification method is applied to the MI motion data for detecting human action.

**MI system setup**. The MI transceivers adopted in the experiments consist of a coil and L-reversed impedance matching network[52]. The matching network is used to maximize the transmission efficiency of the overall system[52]. The coils are identical, air-cored, single layer copper with 5 cm radius, 10 AWG wire diameter, and the user can wear them as accessories. The coil's radius can change depending on the size of the body part that they are designed to be placed around. The source and load impedances are 50 Ω, and the resonance frequency is 13.56 MHz. As the operating frequency is lower than 30 MHz, the human body effect is neglected[33], and the effect of the background medium is considered to be the same as that of air. The reversed L-matching networks consist of a series inductor of 5380 nH and a parallel capacitor of 600 pF.

**Synthetic MI motion data**. In this study, we have synthesized MI motion data to evaluate the proposed MI-HAR system capability in motion detection. The circuit model of the MI system (see Methods section) is used to calculate the forward voltage gain, which is the scaled version of the received MI signal. As the pattern is the same, we used the generated voltage gain patterns of the system as the input features for the detection algorithm. Figure 1 shows the measured and simulated forward voltage gain of two coils during their movement. Since the distance and misalignment between two coils are required as inputs for the

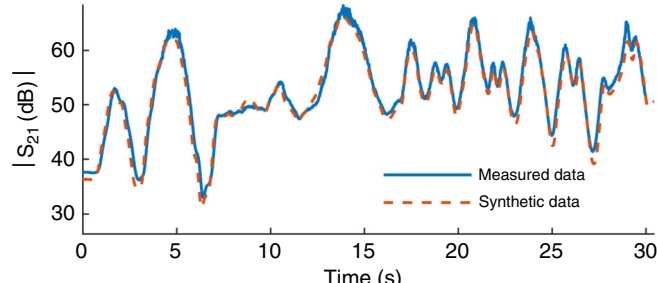

**Fig. 1 Measured vs synthetic magnetic induction (MI) data.** The measured and simulated voltage gain of two MI coils during arbitrary movement, such that both relative alignment and location of coils vary.

model, their location and alignment are captured using video object tracking (see Methods section). Results show that the simulated signal is consistent with the measured data, which is an indication of a valid model for generating time-series MI data. We have performed experiments for 20 different motions that involve both geo-translation and misalignment of coils. The average normalized root-mean-squared error (NRMSE) of the synthesized and measured $S_{21}$ for these experiments is less than 10.3%. The reported NRSME not only takes into account model error but also includes the error associated with the motion tracking algorithm using video and vector network analyzer (VNA) measurements.

To synthesize MI motion data during different human actions, we considered a receiver and eight transmitter coils around the torso, hands, arms, legs, and thighs, respectively (see Methods section). For spatial translation and rotation of human body bones, 3D motion capture (MoCap) datasets are employed. Each pair of markers placed at the joints can define a bone. Hence, the location and alignment of MI coils placed around the body parts can be derived and provided as inputs to the model for synthesizing the corresponding MI motion data. Two publicly available experimental datasets: Biological Motion Library (BML)[53] with 4 activities and Berkeley Multimodal Human Action Database (MHAD)[54] with 11 activities are used here. A brief description of these datasets is presented in the Methods section. The generated synthetic forward voltage gain of the MI transceivers corresponds to these datasets is presented in Fig. 2. A point to consider is that we have extended the single-transmitter/single-receiver model to a multi-transmitter/single-receiver scenario, assuming the interferences such as cross-coupling between coils are negligible, because the interference mitigation techniques such as time-division multiplexing[55] or frequency splitting[56] can be applied to reduce or ideally eliminate interference between inductive systems. Moreover, interference protocols (e.g., RFID interference protocols) can control communication between transceivers while preventing their interference with one other. Therefore, the model can provide a reasonably accurate estimation of multi-coil system performance.

**Performance**. Tracking the motion of body parts during physical activity is critical in characterizing an individual's movement, and collecting data that provide a more accurate representation of these motions results in better activity detection. The MI signals express a strong relationship with the geo-translation of body segments since the system gain is directly affected by distance and misalignment between coils. Distributing more coils around the human body provides comprehensive information about the user's body movements and results in a better distinction between similar actions. We used the MHAD dataset to compare the capability of the MI signal and accelerometer data in estimating

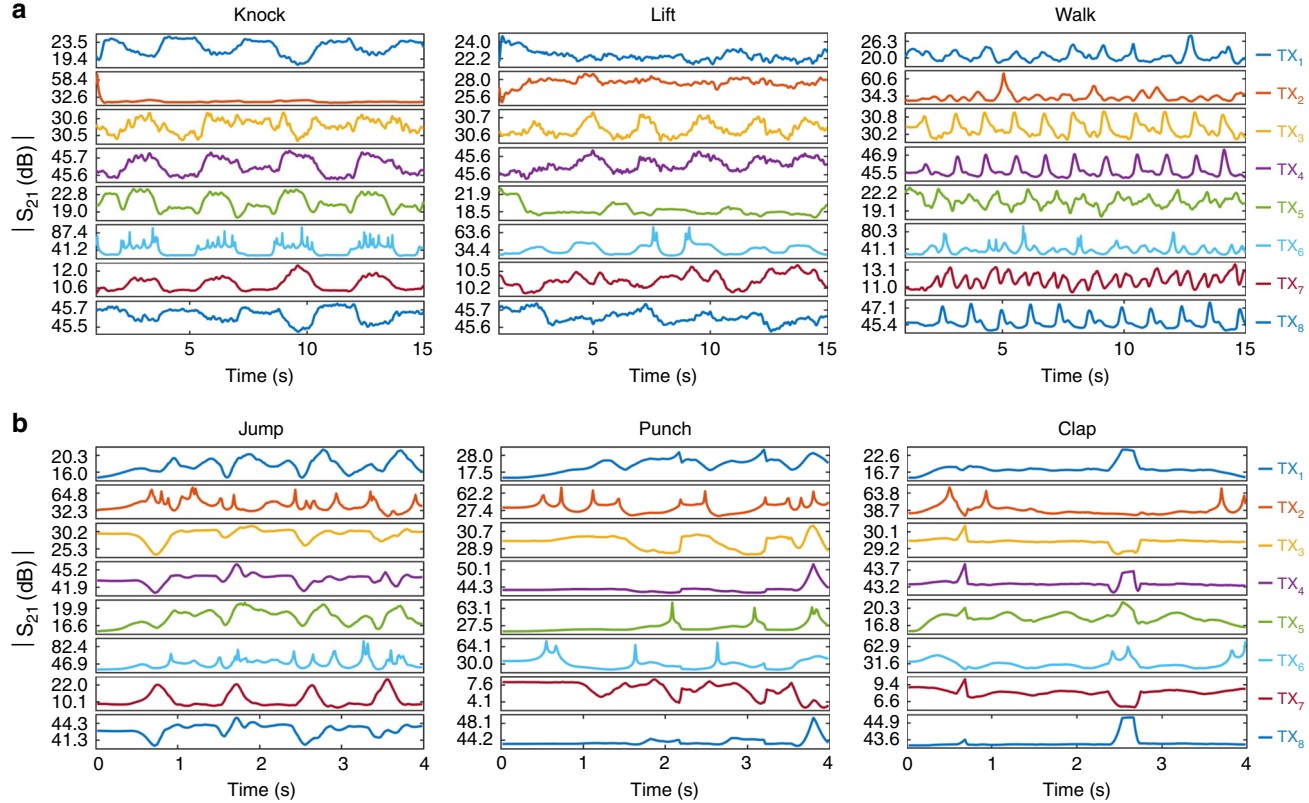

**Fig. 2 Synthetic magnetic induction (MI) motion data.** The forward voltage gain $S_{21}$ between the receiver (RX) and transmitters (TX$_1$–TX$_8$) are generated using the proposed MI model and the human motion data captured for different activities in two datasets: **a** Biological Motion Library (BML) and **b** Berkeley Multimodal Human Action Database (MHAD).

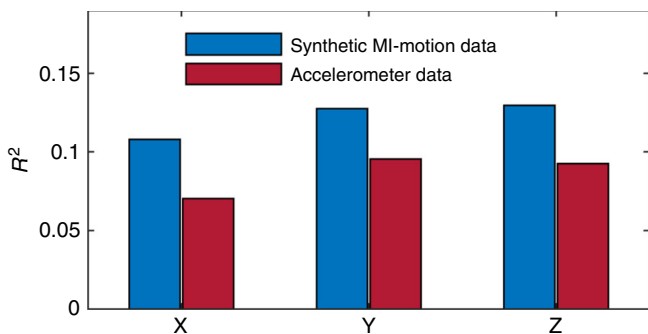

**Fig. 3 Average $R^2$ between XYZ of each target point and data of its corresponding accelerometer and magnetic induction (MI) transceiver.** The $R^2$ reports the similarity between two sets of data by a number between zero and one, where a higher number shows a stronger relationship between two datasets.

**Table 1 Performance summary.**

| Classifier | Overall accuracy | Average precision | Average recall | F1 score |
|---|---|---|---|---|
| BML | | | | |
| SVM | 83.5% | 85.4% | 83.5% | 0.84 |
| KNN | 79.4% | 80.0% | 79.4% | 0.8 |
| Decision Trees | 77.3% | 77.1% | 77.1% | 0.77 |
| Random Forest | 86.6% | 86.5% | 86.5% | 0.86 |
| Logistic regression | 83.5% | 86.2% | 83.6% | 0.85 |
| Deep LSTM | 87.0% | 86.7% | 87.0% | 0.87 |
| MHAD | | | | |
| SVM | 96.4% | 96.6% | 96.4% | 0.96 |
| KNN | 90.3% | 91.1% | 90.3% | 0.91 |
| Decision Trees | 81.2% | 82.3% | 81.2% | 0.82 |
| Random Forest | 90.9% | 91.8% | 90.9% | 0.91 |
| Logistic regression | 90.9% | 91.3% | 90.9% | 0.91 |
| Deep LSTM | 98.9% | 98.9% | 98.9% | 0.99 |

The result of classification models using generated synthetic magnetic induction (MI) motion data of different datasets.

the location of a body part during physical activity. The accelerometer is considered here as a benchmark because it is the most frequently used wearable sensor modality for human activity monitoring. Six markers placed close to the accelerometers are considered as target points. Then the similarity between the 3D location of each target point and data of its corresponding accelerometer and MI transceiver is calculated. We used $R^2$ as the similarity metric, and the average values over the whole dataset are presented in Fig. 3. The results show that, on average, the MI signal has a stronger relationship with the 3D location of markers compared to the accelerometer data. This characteristic can be useful not only in classifying human activities but also in reconstructing the motion trajectories of body segments. Many

studies have adopted IMUs to reconstruct the trajectories of movements for motion analysis in different applications. Examples include handwritten digit recognition[57], monitoring trunk kinematics during standing up to sitting down[58], and tracking the motion of body parts on patients who have been affected by neurological conditions for rehabilitation purposes[59]. In inertial sensor-based recognition systems, the velocity and positions are computed indirectly by the integration over sensor

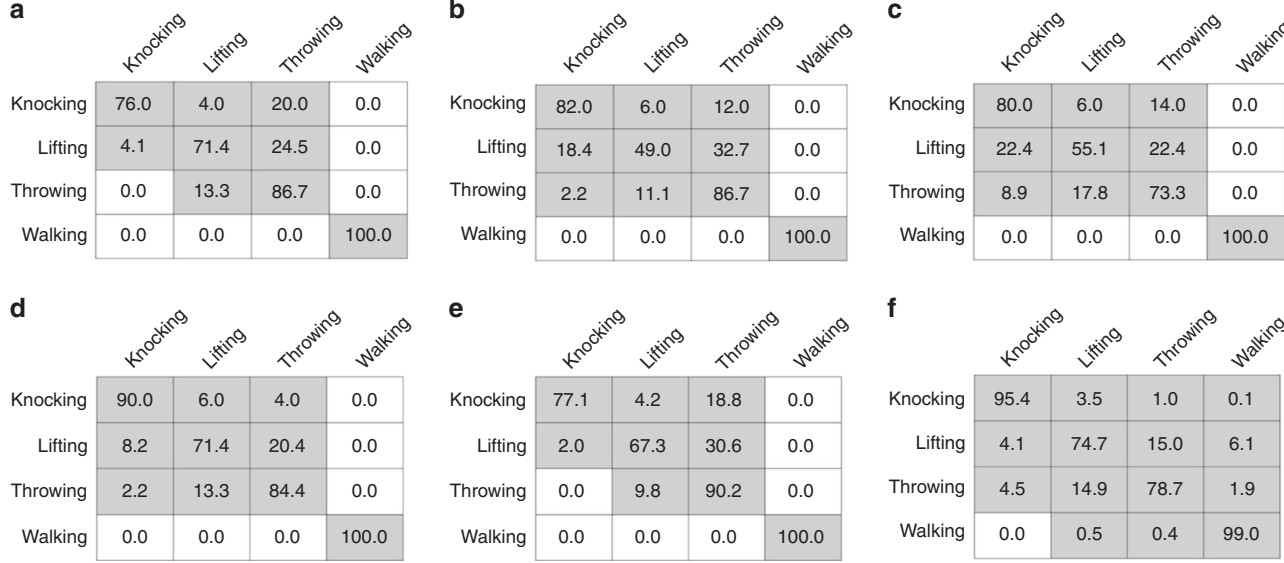

**Fig. 4 Confusion matrix for the validation set corresponding to the Biological Motion Library (BML).** The results correspond to **a** support vector machines (SVM), **b** K-nearest neighbors (KNN), **c** decision trees (DT), **d** random forests (RF), **e** logistic regression (LR), and **f** deep long short-term memory (LSTM) recurrent neural network (RNN) classifiers. The rows and columns represent the percentage of true activity labels and the predicted activity labels, respectively.

measurements. It makes the estimation errors caused by the intrinsic noise/drift grow unbounded with time. For example, the average displacement error of Xsens IMU after 1 min is about 152 m[59]. On the other hand, the MI motion signal is directly affected by the location and orientation of coils. As a result, the trajectory reconstruction using MI signals does not require integration over measured data, which removes the problem of the cumulative error.

To assess the performance of the proposed MI-HAR system in recognizing human activities, we implemented deep recurrent neural networks (RNNs) based on long short-term memory (LSTM) units due to their strong performance in human activity detection, and their capability in learning complex representations of the motion data[60,61]. We compared the results of this method with several commonly used classifiers for activity detection using generated synthetic MI motion data. Table 1 summarizes the performance results of LSTM with methods including support vector machines (SVM), K-nearest neighbors (KNN), decision trees (DT), random forests (RF), and logistic regression (LR). The confusion matrix of each classification method on BML and MHAD datasets are also presented in Figs. 4 and 5, respectively. The results are compared to other previously introduced methods using different modalities for activity detection. We employed accuracy as an evaluation metric for comparison, as datasets used in this paper are balanced and have an equal number of samples for each activity. The results presented in ref. [62] show that SVM and Multi-Task Conditional Restricted Boltzmann Machines (MT-CRBMs) classifiers have achieved an accuracy of 41.3% and 54.5% using BML motion capture data, respectively. For the MHAD dataset[63], has reported an accuracy of 98% by applying SVM on accelerometer data. The random forest classifier has also achieved an accuracy of 96% and 68.2% using MHAD motion capture and audio data[64]. The accuracy of LSTM using camera RGB image for human activity classification is stated as 92.4%[65]. Our results indicate that the deep LSTM model with optimum hyperparameters outperforms other classifiers by a considerable margin on the generated synthetic MI motion data. The recurrent neural networks can capture sequential and time dependencies between input data that

results in a strong performance. The LSTM cells let the model capture even longer dependencies compared to vanilla cells. A deep architecture with an optimal number of layers enables the neural network to extract useful discriminative features from the set of input data and to improve the performance of the model. It should be noted that the datasets used in this paper are diverse, which proves the classifier models are valid for a broad range of activity recognition tasks. Moreover, the actions recorded in the BML dataset, including knocking, lifting, and throwing, are very similar as only one hand is moving. The same movement of human body parts in these activities makes it difficult to distinguish and categorize them. Despite these challenges, the deep LSTM model has achieved high accuracy, and it indicates that the recurrent model is capable of classifying human actions by using MI motion signals.

## Discussion

HAR is a powerful technology with a wide range of applications such as healthcare, rehabilitation, sports training, and senior monitoring. We proposed a new wearable-based HAR system using MI for motion capture and wireless signal transmission. This method can tackle existing issues with conventional HAR systems in various aspects, including power consumption, the complexity of implementation, and cost. It can also provide a suitable infrastructure for new applications working in harsh environments, such as underwater. The proposed system is a new sensing approach for capturing human motions, which can also be integrated with other monitoring modalities to provide a more comprehensive HAR system.

To show the capability of the MI-HAR system in detecting human movements, we generated synthetic MI motion data received from MI transmitters around the user's body during different activities by the MI system model. As mentioned before, the model used for synthesizing MI motion data does not consider cross-coupling between transmitter coils. However, this cross-coupling is not necessarily destructive and can even provide further information regarding the location and alignment of all coils relative to each other. In this scenario, each received signal is

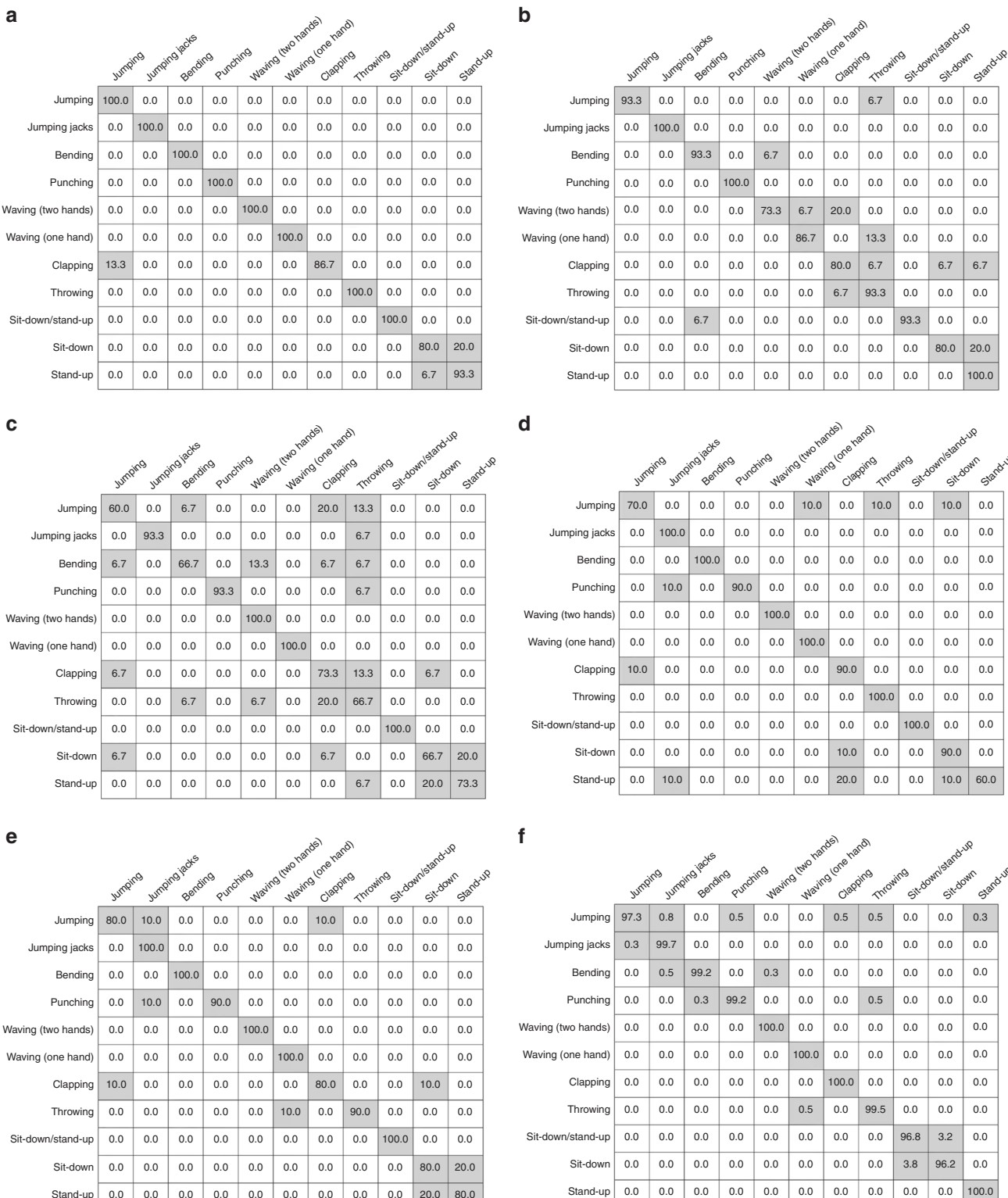

**Fig. 5 Confusion matrix for the validation set corresponding to the Berkeley Multimodal Human Action Database (MHAD).** The results correspond to **a** support vector machines (SVM), **b** K-nearest neighbors (KNN), **c** decision trees (DT), **d** random forests (RF), **e** logistic regression (LR), and **f** deep long short-term memory (LSTM) recurrent neural network (RNN) classifiers. The rows and columns represent the percentage of true activity labels and the predicted activity labels, respectively.

not only a function of the transmitter and receiver coils but also the arrangement of all other coils affects it. Therefore, the movement of even a single body part results in a different signal pattern and can make the system more accurate in detecting actions similar to each other. In the future, we plan to build a realistic deployment-ready prototype of the MI system for capturing MI motion signals during various human activities. Such a system would allow us to perform experiments on real-world MI motion data to demonstrate the accuracy of our method and study the effect of cross-coupling interference on the MI-HAR system. The proposed system can also be integrated with other modalities and monitoring techniques to provide a more comprehensive system for human motion tracking.

We employed several commonly used machine leaning-based classifiers and deep recurrent neural networks for the detection step. We empirically evaluated the proposed MI-HAR system by conducting experiments on the generated synthetic MI motion dataset and discussed the outcomes in detail. Experimental results reveal that the proposed deep LSTM model shows outstanding performance compared to other approaches. One of the benefits of using the deep recurrent neural network for sequence classification is that it can support multiple parallel temporal input data from different sensor modalities such as MI sensors, accelerometers, and gyroscopes. The model can learn complex features directly from raw data and map them to activities. It removes the need for manual feature engineering by experts while it achieves a comparable performance to models with the feature handcrafting step. Besides, the neural network model enables an interactive learning system when the user provides training data even after the initial training step. It allows the user to fine-tune a pre-trained neural network model with their personal data. However, the neural network complexity should be assessed where models have to be implemented in embedded systems with limited processing capability. It highlights the importance of trade-off between computational cost and detection accuracy to ensure real-time feedback.

## Methods

**Theoretic circuit modeling of the MI system.** The MI system consisting of two coils can be modeled as a two-port network shown in Fig. 6. Coils are attached to impedance matching networks, called input and output matching networks, to maximize the transmission efficiency of the overall system[52]. The closed-form expressions of these circuit parameters are reported in ref. [33] to facilitate performance analysis of the MI-based communication system around the human body. The model is validated by simulations and measurements performed for various coils in different locations and alignments relative to each other[33]. The average error of all experiments compared with the simulated signal attenuation results is lower than 10% in the frequency range below 30 MHz. The more advanced version of the expressions without any simplification is also calculated and reported in this work.

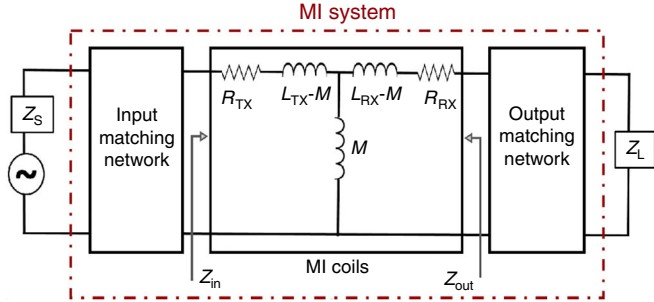

**Fig. 6 Equivalent two-port network model of magnetic induction (MI) system.** The model is integrated with input and output matching networks, and $M$, $L_{TX}$, $R_{TX}$, $L_{RX}$, and $R_{RX}$ are the mutual inductance between coils, inductance, and resistance of transmitter and receiver coils, respectively.

Assume that the transmitter coil with number of turns $N_{TX}$, area $S_{TX}$, and current $I_{TX}$ is centered at $\mathbf{C}_{TX}$, and its surface normal is $\hat{\mathbf{n}}_{TX}$. The receiver coil with number of turns $N_{RX}$ and area $S_{RX}$ is centered at $\mathbf{C}_{RX}$, and its surface normal is $\hat{\mathbf{n}}_{RX}$. The mutual inductance between the coils in a linear, homogeneous, isotropic background medium with permeability $\mu$ and complex propagation constant $\gamma$ can be calculated from $M = \frac{\mu N_{TX}}{I_{TX}} \int_{S_{RX}} \mathbf{H}_{TX} . d\mathbf{S}_{RX}$ [33,66]. By using the exact expressions for the magnetic field generated by the TX coil $\mathbf{H}_{TX}$ and applying a procedure similar to ref. [33], one can derive the mutual inductance without any simplification as follows:

$$M = \frac{\mu N_{TX} N_{TX} S_{TX}}{4\pi} \int_{\phi'=0}^{2\pi} \int_{\rho'=0}^{a_{RX}} \rho \, d\phi d\rho$$
$$[-\rho^2 \cos\alpha - \rho \sin\phi \,(1+\cos^2\alpha)(\mathbf{c}_{rx}.\hat{\mathbf{y}})$$
$$- \rho \sin\phi \sin\alpha \cos\alpha (\mathbf{c}_{rx}.\hat{\mathbf{z}}) - 2\rho \cos\phi \cos\alpha (\mathbf{c}_{rx}.\hat{\mathbf{x}})$$
$$- \cos\alpha(\mathbf{c}_{rx}.\hat{\mathbf{x}})^2 - \cos\alpha(\mathbf{c}_{rx}.\hat{\mathbf{y}})^2 - \sin\alpha(\mathbf{c}_{rx}.\hat{\mathbf{y}})(\mathbf{c}_{rx}.\hat{\mathbf{z}})].$$
$$\mathcal{R}\{\frac{1+\gamma r + \gamma^2 r^2}{r^5} e^{-\gamma r}\}$$
$$+ [\cos\alpha(\mathbf{c}_{rx}.\hat{\mathbf{z}})^2 - \sin\alpha(\mathbf{c}_{rx}.\hat{\mathbf{z}})(\mathbf{c}_{rx}.\hat{\mathbf{y}})$$
$$- \rho \sin\phi \sin^2\alpha(\mathbf{c}_{rx}.\hat{\mathbf{y}}) + \rho \, \sin\phi \, \cos\alpha \, \sin\alpha(\mathbf{c}_{rx}.\hat{\mathbf{z}})].$$
$$\mathcal{R}\{\frac{1+\gamma r}{r^5} e^{-\gamma r}\}], \tag{1}$$

where $r$ is the distance between the origin and the observation point and can be defined in the cylindrical coordinates as follows:

$$r = r(\rho, \phi)$$
$$= [\rho^2 + (\mathbf{c}_{rx}.\hat{\mathbf{x}})^2 + (\mathbf{c}_{rx}.\hat{\mathbf{y}})^2 + (\mathbf{c}_{rx}.\hat{\mathbf{z}})^2$$
$$+ 2\rho \, \sin\phi \, [\cos\alpha(\mathbf{c}_{rx}.\hat{\mathbf{y}}) + \sin\alpha(\mathbf{c}_{rx}.\hat{\mathbf{z}})]$$
$$+ 2\rho \cos\phi(\mathbf{c}_{rx}.\hat{\mathbf{x}})]^{1/2}, \tag{2}$$

The parameters used in the above expressions are calculated from location and alignment of TX/RX coils as follows:

$$\alpha = \tan^{-1}\left(-\frac{\hat{\mathbf{n}}_{rx}.\hat{\mathbf{y}}}{\hat{\mathbf{n}}_{rx}.\hat{\mathbf{z}}}\right), \tag{3}$$

$$\hat{\mathbf{n}}_{rx} = \mathbf{R}_z(\theta_z) \, \mathbf{R}_y(\theta_y) \, \mathbf{R}_x(\theta_x) \, \hat{\mathbf{n}}_{RX}, \tag{4}$$

$$\mathbf{C}_{rx} = \mathbf{R}_z(\theta_z) \, \mathbf{R}_y(\theta_y) \, \mathbf{R}_x(\theta_x)(\mathbf{C}_{RX} - \mathbf{C}_{TX}), \tag{5}$$

$$\theta_x = \tan^{-1}\left(\frac{\hat{\mathbf{n}}_{TX}.\hat{\mathbf{y}}}{\hat{\mathbf{n}}_{TX}.\hat{\mathbf{z}}}\right), \tag{6}$$

$$\theta_y = \tan^{-1}\left(-\frac{\hat{\mathbf{n}}_{TX}.\hat{\mathbf{x}}}{\sqrt{(\hat{\mathbf{n}}_{TX}.\hat{\mathbf{y}})^2 + (\hat{\mathbf{n}}_{TX}.\hat{\mathbf{z}})^2}}\right), \tag{7}$$

$$\theta_z = \tan^{-1}\left(\frac{(\hat{\mathbf{n}}_{RX}.\hat{\mathbf{x}})[(\hat{\mathbf{n}}_{TX}.\hat{\mathbf{y}})^2 + (\hat{\mathbf{n}}_{TX}.\hat{\mathbf{z}})^2]}{(\hat{\mathbf{n}}_{RX}.\hat{\mathbf{y}})(\hat{\mathbf{n}}_{TX}.\hat{\mathbf{z}}) - (\hat{\mathbf{n}}_{TX}.\hat{\mathbf{y}})(\hat{\mathbf{n}}_{RX}.\hat{\mathbf{z}})} \right.$$
$$\left. - \frac{(\hat{\mathbf{n}}_{TX}.\hat{\mathbf{x}})[(\hat{\mathbf{n}}_{TX}.\hat{\mathbf{y}})^2(\hat{\mathbf{n}}_{RX}.\hat{\mathbf{y}}) - (\hat{\mathbf{n}}_{TX}.\hat{\mathbf{z}})(\hat{\mathbf{n}}_{RX}.\hat{\mathbf{z}})]}{(\hat{\mathbf{n}}_{RX}.\hat{\mathbf{y}})(\hat{\mathbf{n}}_{TX}.\hat{\mathbf{z}}) - (\hat{\mathbf{n}}_{TX}.\hat{\mathbf{y}})(\hat{\mathbf{n}}_{RX}.\hat{\mathbf{z}})}\right), \tag{8}$$

where $\mathbf{R}_x(\theta_x)$, $\mathbf{R}_y(\theta_y)$, $\mathbf{R}_z(\theta_z)$ are rotation matrices that rotate vectors by an angle $\theta_x$, $\theta_y$, $\theta_z$ about the x-, y-, or z-axis using the right-hand rule.

The self-inductance and resistance, which comprises DC resistivity, skin depth $\delta_w$, and proximity effects, of a coil with radius $a$, length $b$, number of turns $N$, circular cross-section wire, core-material permeability $\mu$, wire diameter $\phi_w$, and wire resistivity of $\rho_w$ can be expressed as follows[33,67]:

$$L = \mu \, a \, N^2 \left[\log\frac{8a}{b} - \frac{1}{2} + \frac{b^2}{32 a^2}\left(\log\frac{8a}{b} + \frac{1}{4}\right)\right.$$
$$- \frac{b^4}{1024 a^4}\left(\log\frac{8a}{b} - \frac{2}{3}\right) + \frac{10 b^6}{131072 a^6}\left(\log\frac{8a}{b} - \frac{109}{120}\right)$$
$$\left. - \frac{35 b^8}{4194304 a^8}\left(\log\frac{8a}{b} - \frac{431}{420}\right)\right], \tag{9}$$

$$R = \begin{cases} \frac{2 a N \rho_w}{\delta_w (\phi_w - \delta_w)} & \text{if } \phi_w < \delta_w \\ \frac{8 a N \rho_w}{\phi_w^2} & \text{if } \phi_w \geq \delta_w \end{cases}. \tag{10}$$

There are multiple parameters for analyzing and determining the performance of a two-port network. As the proposed MI-based communication system is a cascaded connection of two-port networks, the ABCD parameters are the best candidate. These parameters, which are also known as transmission, chain, or cascade parameters, relate the input current and voltage at port-1 to the output. The ABCD parameters of the MI system are equivalent to the product of ABCD matrices corresponding to the input matching network, the MI system, and the output matching network, accordingly[68].

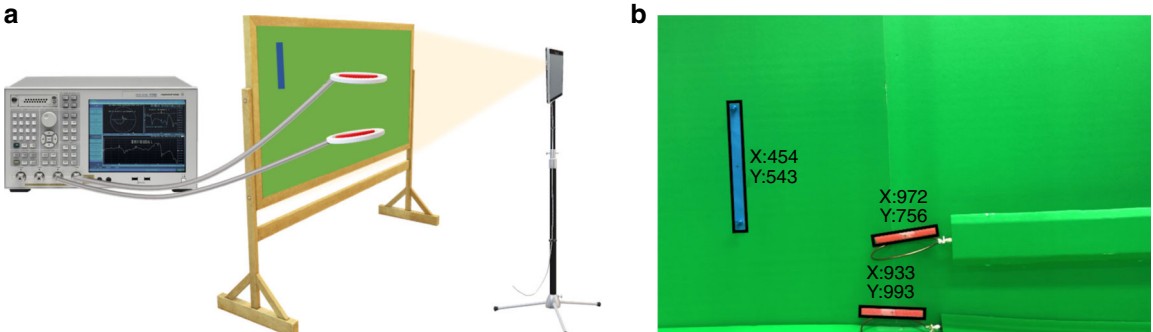

**Fig. 7 Measurements. a** Schematic representation of measurement setup. **b** The camera frame sample after video processing for object tracking and extracting red and blue markers attached to coils and the fixed-length calibration label, respectively.

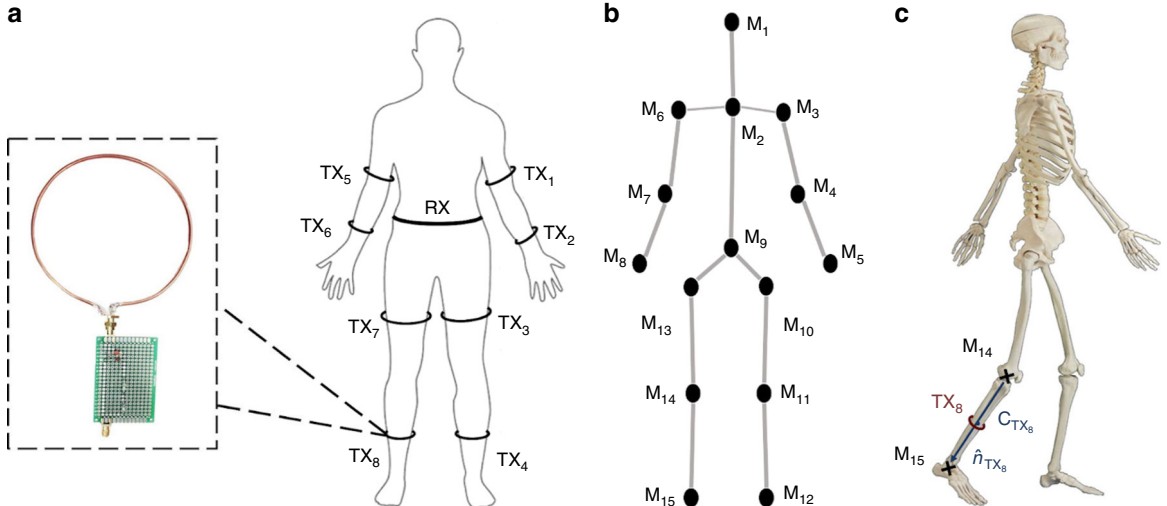

**Fig. 8 Magnetic induction (MI)-based communication system. a** Location of MI transceivers (TX$_i$, RX) on the human body and the laboratory version of an MI transceiver. **b** Location of markers (M$_j$) on the body. The marker pairs of (2,9), (3,4), (4,5), (10,11), (11,12), (6,7), (7,8), (13,14), (14,15) define two ends of the torso, left arm, left hand, left thigh, left leg, right arm, right hand, right thigh, and right leg, respectively. Consequently, these pairs can be utilized to calculate the location of coils (C$_{TX_i}$, C$_{RX}$) and their alignment ($\hat{n}_{TX_i}$, $\hat{n}_{RX}$). **c** The center and alignment of a bone and its corresponding coil can be calculated using markers locations.

The scattering matrix $S$ is another set of two-port parameters defined in terms of incident and reflected waves at ports. One of the matrix elements is forward voltage gain $S_{21}$, which shows the voltage of the network at port two divided by the voltage at port-1. Converting the ABCD parameters to S-parameters, the forward voltage gain of the MI system can be determined as follows[68]:

$$S_{21} = \frac{2\sqrt{\mathcal{R}\{Z_S\}\ \mathcal{R}\{Z_L\}}}{A\ Z_L + B + C\ Z_S\ Z_L + D\ Z_S},\qquad(11)$$

where $A$, $B$, $C$, $D$ are the ABCD parameters of the overall MI system including the MI coils and the matching circuits.

**Measurement.** The forward voltage gain of two coils is measured for 30 s via a VNA with 1800 points resolution. The corresponding synthetic $S_{21}$ is also generated by using the system model for comparison. All parameters of the model are predefined based on the MI system setup except the distance and misalignment between coils, which are variable during the movement. Hence, two coils are labeled with red markers and placed in front of a green screen. The motion of coils is captured via an iPhone's built-in camera with 30 fps, and the videos are processed offline to extract markers, their center, and alignment, as shown in Fig. 7. Since only one camera is used, without loss of generality, coils only move in 2D such that the camera can capture their motion. The extracted pixel-wise movement of coils is then converted to the spatial translation using a predefined length 'calibration label'. The ratio of the calibration label's length to its size extracted from video provides a meter to pixel ratio. As the camera is fixed during the experiment, this ratio remains constant for all frames of the video. The recorded distance between coils covers up to 60 cm range. The generated synthetic MI data are synchronized with measured data by minimizing the NRMSE. The code used to track coils and calculate the forward voltage gain of the system based on the circuit model reported in this work is implemented in MATLAB.

**Simulation.** Figure 8a depicts the location of coils considered around the human body for generating synthetic MI motion data. The location of markers required to track coils motion is also displayed in Fig. 8b. Assuming that the coils are located at the midpoint of bones, we can calculate their center by averaging the location of corresponding paired markers. For example, Fig. 8c shows the right leg, its corresponding transmitter coil, and markers. The center of the transmitter coil TX$_8$ can be calculated as $c_{TX_8} = (M_{14} + M_{15})/2$. The coils are around the human bones, which indicates that the alignment of the line passing through the markers is the same as the surface normal of its corresponding coil. Therefore, the surface normal of the transmitter TX$_8$ can be written as $\hat{n}_{TX_8} = (M_{15} - M_{14})/|M_{15} - M_{14}|$. The code used to calculate the forward voltage gain of the system based on the circuit model reported in this work is implemented in MATLAB.

**Datasets of human activities.** The experimental datasets considered in this work contain diverse movement data to verify the applicability of MI-HAR in detecting a wide range of activities. The BML dataset[53] contains a full-body movement dataset for walking, knocking, lifting, and throwing performed by 15 male and 15 female actors in a neutral, angry, happy, and sad style. The dataset is balanced and has the same number of records performed by actors for each action. The total number of samples is 1028, with a sampling rate of 60 Hz. For walking action, the data are captured for 30 s of walking in a triangle turning rightward (clockwise), and turning leftward (counterclockwise). For the knocking, lifting, and throwing actions, five repetitions of a single action unit are obtained for each data record, which is approximately 20 s in duration. The MHAD dataset[54] contains the data for jumping, jumping jacks, bending, punching, waving two hands, waving one hand, clapping, throwing, sit down/stand up, sit down, stand up. The number of records for each action is the same, and each action is performed by seven male and five female subjects, yielding about 659 data sequences. Except for sitting down, stand up, and throwing, all records include five repetitions of a single action. The

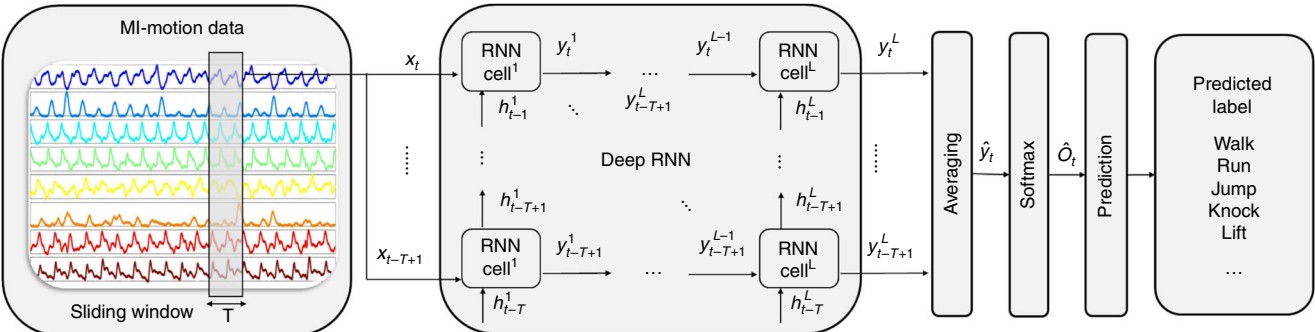

**Fig. 9 Architecture of deep recurrent neural network (RNN).** The set of magnetic induction (MI) signals observed by the coils at time $t$ is considered as the input vector $\mathbf{x}_t$. A time window of 1 s (T = 1 s) is sliding over the data with 0.5 s overlap, and feeding the truncated subsequences of input data within the window to the batch normalization layer. Then the normalized input data ($\mathbf{x}_{t-T+1}, \ldots, \mathbf{x}_{t-1}, \mathbf{x}_t$) is fetched to the deep long short-term memory (LSTM) model. The network outputs sequences of vectors ($\mathbf{y}_{t-T+1}^L, \ldots, \mathbf{y}_{t-1}^L, \mathbf{y}_t^L$), where each output vector shows the prediction score of its corresponding input sample. Assuming the input signals are sequenced to $N$ samples, the overall score of the entire window can be calculated by averaging all of the scores within the window into a single prediction vector of scores $\hat{\mathbf{y}}_t$[70]. Then the prediction scores are converted into class membership probabilities $\hat{\mathbf{O}}_t$ by applying a softmax layer. The predicted class membership probability vector contains the probability of every class generated by our model. Then the most probable class is selected as the predicted activity label for the given input data within the time window.

approximate recording length of activities varies from 2 to 15 s. The dataset consists of data from four microphones with a sampling rate of 48 kHz; six accelerometers fixed on wrists, hips, and ankles with a sampling rate of 30 Hz, the optical motion capture system with a sampling rate of 480 Hz, cameras with a sampling rate of 22 Hz, and depth sensors with a sampling rate of 30 Hz. In our experiments, we used the down-sampled MoCap data to 60 Hz.

**Data preprocessing**. In our experiments, we have used the magnitude of MI signals as input for the classifiers. Data samples are processed before fetching into the classification models. The processing methods are implemented using Python 3.6. For data cleaning, the missing values are substituted with previous non-missing values, and a 5-point quadratic (order 4) polynomial Savitzky-Golay filter is applied for denoising. Then the baseline offset is removed from time-series data. In the MHAD dataset, 3% of the signals are removed from the end of each data sample as the reported experiments show improvement in the accuracy[63].

**Classification**. The classifier models are implemented using Python 3.6. They are trained and evaluated on the generated synthetic motion datasets of eight bones using the leave-one-subject-out cross-validation (LOSO-CV) method. For the experiments on the BML and MHAD dataset, respectively, six and two subjects are used for validation and the rest for training.

Machine learning-based classifiers: The machine learning-based classifiers are implemented using python library Sklearn[69]. The multi-class models are non-linear SVM with a polynomial kernel, KNN, decision trees, random forests, and logistic regression. We used the bag-of-words (BoW) representation to characterize the time-series data with different lengths. First, the synthetic MI motion data are divided into fixed-length segments of 1 second using the sliding window technique with 0.8 second overlap. Attributes are then computed for the time domain, frequency domain, and time-frequency domain of each window segment. Frequency domain and time-frequency domain representations of the signal are calculated by the fast Fourier transform (FFT), and single-level discrete Wavelet transform (DWT) based on the Daubechies2 wavelet filter, respectively. The attributes considered here are extremes, mean, median, standard deviation, lower quartile, upper quartile, skewness, kurtosis, and the correlation between each pair of signals. As each action is associated with eight data samples, the resulting feature vector for each segment is generated by the concentration of eight feature sets. Features are also scaled using the min-max scaling method to bound values in the range of 0–1. The scaling makes the weight of all features equal in the process of classification. Next, the feature vectors from the training data are clustered using k-means clustering to define a codebook that contains the cluster centers, which are called codewords. Then, each window segment is assigned the closest codeword, and a time-series is represented as a histogram of codewords. The bag-of-words representations of synthetic MI motion data are used as inputs for the machine learning-based classification models. In our experiments, we quantized the training data of BML and MHAD datasets to 100 and 20 codewords, respectively.

Recurrent neural network: A schematic diagram of the neural network structure is summarized in Fig. 9. The deep LSTM model is implemented in the TensorFlow framework. We used the mean cross-entropy between the ground truth labels and the predicted class membership probability vector as the loss function, and the network parameters are updated by minimizing this loss function. The model is trained using batch gradient descent with the RMSprop updating rule. In each epoch of training, the entire training set is passed through the neural network model to update the model with an exponentially decaying learning rate. The

dropout regularization technique is also applied to all nodes in the network to avoid overfitting. The dropout keep-probability determines the probability of keeping a node during training. After each epoch, the performance of the model is evaluated on the validation set. We evaluated the influence of several hyperparameters related to the network architecture and learning process using grid-search. These hyperparameters and their range of values explored for tuning during training are: number of layers in the range of {1, 2,3, 5, 10}, number of units in the range of {5, 10, 15, 20, 30, 40, 50}, keep probability in the range of {0.2, 0.5, 0.8, 1}, optimizer decay rate in the range of {0.8, 0.85, 0.9, 0.95, 0.98}, optimizer momentum in the range of {0, $10^{-3}$, $10^{-2}$, $10^{-1}$}, initial learning rate in the range of {$10^{-3}$, $10^{-2}$, $10^{-1}$}, exponential decay rate in the range of {0.85, 0.9, 0.95, 0.98}, and exponential decay step in the range of {50, 100, 200, 300}. We implemented a five-layer network with 20 and 40 units for BML and MHAD datasets, respectively. Both datasets are trained with the optimizer decay rate of 0.95, the initial learning rate of 0.01, the exponential decay rate of 0.98, exponential decay step of 100, and keep the probability of 0.8.

## Data availability
The data that support the findings of this study can be reproduced using the codes developed in this work and are also available on Figshare (https://doi.org/10.6084/m9.figshare.c.4844517). The raw data that our synthetic MI motion data were derived from are available in the public domain: BML dataset (http://paco.psy.gla.ac.uk); MHAD dataset (http://tele-immersion.citris-uc.org/berkeley_mhad).

## Code availability
Computer code supporting the findings of this study are available on GitHub: synthesizing MI data (https://github.com/negargolestani/Synthesize_MI_data); Activity detection (https://github.com/negargolestani/Activity_Detection).

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

## Acknowledgements

This work was supported in part by a grant (80NSSC17K0283) from the National Aeronautics and Space Administration (NASA) Earth Science Technology Office (ESTO) Advanced Information Systems Technology (AIST) program.

## Author contributions

N.G. was the main contributor to this work and was responsible for developing and implementing the methods, data generation, and analysis; M.M. supervised the research.

## Competing interests

The authors declare no competing interests.
