## [Peer Review File · Nature Communications]

Reviewers' comments:

Reviewer #1 (Remarks to the Author):

The paper addresses a very important and interesting problem of human activity recognition having a lot of useful applications. The authors correctly state that HAR is still an open scientific problem. The paper introduces a very interesting and novel idea of using a wireless system based on magnetic induction (MI). The signals obtained using this system are then processed by modern and sophisticated machine learning algorithms, i.e., deep recurrent neural networks. The results obtained on synthetically created dataset for four categories of physical activities, namely walk, knock, lift, and throw are convincing.

The idea of using magnetic induction for the acquisition of signals to be processed is very impressive, novel and definitely worth further scientific investigations. However, do the experiments performed on a synthetically created dataset for just four categories of physical activities really show the evidence for the usefulness of this methodology? Is it not too early for this kind of statement? Moreover, the subject-dependent evaluation scheme is also sub-optimal. I think a leave-one-subject-out-cross-validation might be more suitable than the regular 5-fold cross validation which was used in their experiments.

Summarising, I would suggest to the authors to follow this fascinating research by extending the experimental setup significantly.

Reviewer #2 (Remarks to the Author):

The manuscript presents a technique based on magnetic induction principles to track the movements performed by subjects wearing components (i.e. coils) of the proposed system. Data collected in such manner would be then processed using machine learning techniques to detect the motor tasks performed by subjects. The results presented in the manuscript are based on simulations derived using an existing dataset made publicly available by the Perception Action and Cognition Laboratory at the University of Glasgow. The approach is of interest, but – unfortunately - the work summarized in this manuscript is marked by major shortcomings. First of all, it must be emphasized that the data analyzed is synthetic data. In other words, the authors took an existing dataset of movement patterns associated with the performance of various motor tasks (e.g. walking,

lifting, ...) and simulated the use of the proposed system based on the above-mentioned magnetic induction principles. The model utilized to synthesize the data is a “theoretical” model that appears to neglect several sources of interference. It is not clear if the method utilized by the authors to synthesize the data takes properly into account the potential interference among coils positioned on the body and the presence of additional ferromagnetic material close to the subject undergoing monitoring. This is a source of concern because it is known that commercially-available systems based on similar principles have not been successful because of the interference caused by ferromagnetic material near the subject. The analysis of the data is also marked by major shortcomings. For instance, the results appear to be affected by significant overfitting. This is clear if one considers the significant difference between training and test set results when in fact the authors used a 5-fold cross-validation technique. A more appropriate way to characterize the proposed algorithm would have been to test its generalizability using a leave-one-subject out approach. It is very likely that the poor generalizability of the results shown in the manuscript are due to limited size of the training set. The authors should have utilized a machine learning technique suitable for small datasets. Their claim that the proposed approach can be used with a relatively small dataset is, in my opinion, incorrect. In fact, the results appear to suggest otherwise. Another major concern is that the authors do not show any data concerning the accuracy of the movement trajectory reconstructions. A potential advantage of the proposed technique (though this is not clearly stated in the manuscript) is the possibility of generating accurate estimates of the position and orientation of body segments via “direct measure” rather than by integrating data collected using IMU’s. Hence, the authors should have provided a detailed characterization of the estimation errors associated with the reconstruction of position and orientation of body segments rather than “jumping” to the detection of different motor tasks. It is concerning that the authors did not emphasize this potential advantage of the proposed method and did not characterize the algorithm accordingly. It is also concerning that the authors made very weak statements to support the potential interest for the proposed system. In fact, they do not provide a fair summary of existing literature. A better summary would have acknowledged the relevance of methods based on IMU’s to reconstruct the trajectory of movements of body segments. It would have also emphasized the limitations of existing techniques and characterized the proposed approach accordingly. Finally, it is concerned that authors used unusual nomenclature (i.e. “external approach”) to refer to the use of ambient sensors and it is also concerning that they pull together ambient sensors and camera-based motion tracking systems.

Reviewer #3 (Remarks to the Author):

Summary:

This paper proposes to take advantage of magnetic induction coils to recognize humans' activity in their daily life. Having two MI coils attached to two different parts of the human body, we can

observe some discerning patterns in different activities by monitoring the changes in the distance and alignments between these MI coils. The idea of proposing a new method of sensing human body's motion is interesting and promising and authors provided some other motivations to use MI coils instead of existing solutions, such as motion sensors and radio signals. The method and its requirements are explained in details. The MI motion data has been synthesized to evaluate the performance of machine learning models in a HAR setting.

Major Comments:

There are some important questions and ambiguities that should be addressed. In the following, I am providing my comments and suggestions:

Authors mentioned some criteria to compare different approaches for HAR, like “power consumption, coverage, accuracy, cost, and quality of service (QoS)”. So, they need to compare MI-based HAR with motion sensor-based HAR in at least one of these. I expected to see, for example, what is the accuracy of detecting walking from a wristband accelerometer in comparison to these MI coils? Especially, for the cost, when we already have several sensors embedded in our smartphones and smartwatches, how much it cost us to add or use this MI coils? Authors say: “monitoring systems should be inexpensive”, but we don’t see a comparison with already existing approaches.

Authors say: “The wearable devices should be small, lightweight, and not hinder the user’s movements, which puts a restriction on the battery size and longevity.” Again, how do you compare your approach that needs to attach MI coils to the human body with other approaches that use embedded sensors on smart devices? In which one the proposed approach beats other technologies?

Authors say: “inertial sensors such as accelerometers, gyroscope, and magnetometer, are attached to the user, and the collected data are transferred wirelessly to the processing node for motion detection.” Or, they say: “It eliminates the need for an extra wireless module and reduces power consumption by combining data collection and wireless signal transmission steps”. This is not clear to me. I couldn’t understand why accelerometers, gyroscope, and magnetometer, in my smartphone, or smartwatch, should send this data to a wireless node when they can process data on-device very well.

Authors say: “the automatic recognition of complex human activities in daily life using multiple wearable sensors is still an open research topic”., and “Although sensor systems can overcome many disadvantages of the external approach, they still face many challenges”. But the references are some surveys from 2012 and 2013. There are many recent studies in this area that show interesting results in HAR using already embedded motion sensors in smart devices.

The set of activities (walk, knock, lift, and throw) is a small set in size and not the most common and challenging activities. For example, I expected you evaluate the performance of some activities that are difficult to distinguish by other approaches, like walking in a flat area with walking downstairs or upstairs.

Overall, the paper has many claims on the supremacy of the MI approach, but we don't see a fair comparison between the MI approach and current approaches like smart devices motion sensors. It is important to clearly know in what exact situation this MI approach beats other approaches. My suggestion is to compare with the refs. [24] or [28].

* Results reported in the Results section is better to be in a table or something more structured. Writing a paragraph full of numbers makes it hard to follow the main findings and compare the results.

* As the techniques and methods for using MI coils to collect human motions data have already been presented in other authors' publications, like refs. [16], [17], [18], and [26], the main novelty of this paper seems to be applying RNNs for activity recognition based on data collected by MI coils. Thus, we expect to see more evaluation of this part of the paper. For example, what if we use a more simple and lightweight ML model, like random forests or if we use a convolutional network than RNNs? We know RNNs give us better accuracy, but the cost of running RNNs should also be considered for real-time applications.

Minor comments:

In the abstract and introduction, what is the exact role of “wireless sensor networks” in recognizing human physical activities? Why we need “wireless sensor networks” when we can simply use an application on the user's smartphone or their smartwatch? Moreover, in practice, where will these "Deep Recurrent Neural Networks" be run? It is very good to have a top-level architecture that explains these details.

It's better to use “average classifications accuracy” than “Success rate”, as it is the common and well-known term in ML literature.

Authors say: "Required memory, security, and privacy of the collected personal data should be guaranteed by the system". I couldn't find any evaluation or discussion to compare the security or privacy concerns of the proposed method with other approaches, like sensor-based one. Do we have some benefits in using MI coils in these specific concerns? It is good to have a brief discussion on these challenges.

"As long as the MI system operates in the induction regime, it can work well in any non-magnetic environment such as around the human body." Please more explain this. Does this mean when the user is using some magnets or something that has some magnets into it, then the system will lose accuracy?

We wish to express our appreciation for your in-depth comments, suggestions, and corrections, which have greatly improved the manuscript. Please find below, in blue, our detailed response to the reviewers' comments. The revisions in the paper are shown with red font.

Reviewer 1

1. *The paper addresses a very important and interesting problem of human activity recognition having a lot of useful applications. The authors correctly state that HAR is still an open scientific problem. The paper introduces a very interesting and novel idea of using a wireless system based on magnetic induction (MI). The signals obtained using this system are then processed by modern and sophisticated machine learning algorithms, i.e., deep recurrent neural networks. The results obtained on synthetically created dataset for four categories of physical activities, namely walk, knock, lift, and throw are convincing.*

RESPONSE: Thank you for your kind feedback.

2. *The idea of using magnetic induction for the acquisition of signals to be processed is very impressive, novel and definitely worth further scientific investigations. However, do the experiments performed on a synthetically created dataset for just four categories of physical activities really show the evidence for the usefulness of this methodology? Is it not too early for this kind of statement? Moreover, the subject-dependent evaluation scheme is also sub-optimal. I think a leave-one subject-out-cross-validation might be more suitable than the regular 5-fold cross validation which was used in their experiments.*

RESPONSE: We have done experimental measurements and updated the paper accordingly (Please see page 3, lines 188-202, Fig. 1). We also added a substantial amount of synthetic data of other activities (pages 3-4, lines, 203-218, Fig 2) and applied several other detection algorithms on them (pages 4-7, lines 271-308, Table 1-2, Figs. 4-5). Moreover, we changed the cross-validation method and used the leave-one-subject-out method.

3. *Summarizing, I would suggest to the authors to follow this fascinating research by extending the experimental setup significantly.*

RESPONSE: We have extended the research to experimental measurements, added more types of activities, and synthesized their data. Then applied different classification methods for detection and reported their results. Please see response to item 2, above, for specific locations where these new analyses and results are presented.

Reviewer 2

1. *The manuscript presents a technique based on magnetic induction principles to track the movements performed by subjects wearing components (i.e. coils) of the proposed system. Data collected in such manner would be then processed using machine learning techniques to detect the motor tasks performed by subjects. The results presented in the manuscript are based on simulations derived using an existing dataset made publicly available by the Perception Action and Cognition Laboratory at the University of Glasgow. The approach is of interest, but – unfortunately - the work summarized in this manuscript is marked by major shortcomings.*

RESPONSE: Thank you for your comments.

2. *First of all, it must be emphasized that the data analyzed is synthetic data. In other words, the authors took an existing dataset of movement patterns associated with the performance of various motor tasks (e.g. walking, lifting, ...) and simulated the use of the proposed system based on the above-mentioned magnetic induction principles. The model utilized to synthesize the data is a “theoretical” model that appears to neglect several sources of interference. It is not clear if the method utilized by the authors to synthesize the data takes properly into account the potential interference among coils positioned on the body and the presence of additional ferromagnetic material close to the subject undergoing monitoring. This is a source of concern because it is known that commercially-available systems based on similar principles have not been successful because of the interference caused by ferromagnetic material near the subject.*

RESPONSE: We have revised the paper accordingly and showed the accuracy of the model in generating synthetic MI-motion data (Please see page 3, lines 188-202, Fig. 1). Then using MoCap data, we synthesized MI-motion data of several additional activities and used the results to show that our conclusions are still valid for a much more expanded data set than originally presented (pages 3-4, lines, 203-218, Fig 2). We also emphasized in the paper that this model is an extended version of a single transmitter / single receiver coil, as we can remove the interference of cross-coupling between transmitters using interference mitigation methods (pages 3-4, lines, 218-229).

3. *The analysis of the data is also marked by major shortcomings. For instance, the results appear to be affected by significant overfitting. This is clear if one considers the significant difference between training and test set results when in fact the authors used a 5-fold cross-validation technique. A more appropriate way to characterize the proposed algorithm would have been to test its generalizability using a leave-one-subject out approach. It is very likely that the poor generalizability of the results shown in the manuscript are due to limited size of the training set. The authors should have utilized a machine learning technique suitable for small datasets. Their claim that the proposed approach can be used with a relatively small dataset is, in my opinion, incorrect. In fact, the results appear to suggest otherwise.*

RESPONSE: We applied other machine learning-based classifiers including SVM, KNN, decision trees (DT), random forests (RF), and logistic regression (LR) for activity detection and also updated the results of the deep learning approach (pages 4-7, lines 271-308, Table 1-2, Figs. 4-5). Moreover, we modified the cross-validation method and used the leave-one-subject-out technique.

4. *Another major concern is that the authors do not show any data concerning the accuracy of the movement trajectory reconstructions. A potential advantage of the proposed technique (though this is not clearly*

stated in the manuscript) is the possibility of generating accurate estimates of the position and orientation of body segments via “direct measure” rather than by integrating data collected using IMU’s. Hence, the authors should have provided a detailed characterization of the estimation errors associated with the reconstruction of position and orientation of body segments rather than “jumping” to the detection of different motor tasks. It is concerning that the authors did not emphasize this potential advantage of the proposed method and did not characterize the algorithm accordingly. It is also concerning that the authors made very weak statements to support the potential interest for the proposed system. In fact, they do not provide a fair summary of existing literature. A better summary would have acknowledged the relevance of methods based on IMU’s to reconstruct the trajectory of movements of body segments. It would have also emphasized the limitations of existing techniques and characterized the proposed approach accordingly.

RESPONSE: Thank you for pointing these out. We included a summary of existing literature on movement trajectory. We explained in the paper that the MI system has an advantage over IMUs for trajectory reconstruction as the signal is directly affected by the distance referenced studies on this topic. In addition, we showed that the MI signal has a stronger relationship with the XYZ location of a body part compared to IMU’s data. We addressed these concerns in the section “Performance”. Please see pages 4-5, lines 230-270, Fig. 3. While the quantitative detection of movement trajectory is out of the scope of the current paper, we agree with the reviewer that it is certainly a problem that can be addressed by the MI-HAR system, and it is indeed the subject of our ongoing work.

5. *Finally, it is concerned that authors used unusual nomenclature (i.e. “external approach”) to refer to the use of ambient sensors and it is also concerning that they pull together ambient sensors and camera-based motion tracking systems.*

RESPONSE: We explained the methods for activity detection in more detail and provided a comprehensive comparison between them in the “Introduction” section. Since accelerometers are the most commonly used sensors for motion monitoring, we have compared them with the proposed method in terms of power, cost, security, etc. (pages 2-3, lines 26-123)

Reviewer 3

Summary:

This paper proposes to take advantage of magnetic induction coils to recognize humans' activity in their daily life. Having two MI coils attached to two different parts of the human body, we can observe some discerning patterns in different activities by monitoring the changes in the distance and alignments between these MI coils. The idea of proposing a new method of sensing human body's motion is interesting and promising and authors provided some other motivations to use MI coils instead of existing solutions, such as motion sensors and radio signals. The method and its requirements are explained in details. The MI motion data has been synthesized to evaluate the performance of machine learning models in a HAR setting.

Major Comments:

There are some important questions and ambiguities that should be addressed. In the following, I am providing my comments and suggestions:

1. *Authors mentioned some criteria to compare different approaches for HAR, like “power consumption, coverage, accuracy, cost, and quality of service (QoS)”. So, they need to compare MI-based HAR with motion sensor-based HAR in at least one of these. I expected to see, for example, what is the accuracy of detecting walking from a wristband accelerometer in comparison to these MI coils?*

RESPONSE: We provided a comparison with other state-of-the-art methods using different modalities for activity detection, including accelerometer data. Table 2 shows the results reported on the datasets (MHAD) we used in our experiments. We also provided a comprehensive introduction to the MI system and its advantages over other approaches (Please see pages 2-3, lines 26-123). We explained in detail what are the existing issues of available activity detection systems and referenced many recent studies in this area.

2. *Especially, for the cost, when we already have several sensors embedded in our smartphones and smartwatches, how much it cost us to add or use this MI coils? Authors say: “monitoring systems should be inexpensive”, but we don’t see a comparison with already existing approaches. Authors say: “The wearable devices should be small, lightweight, and not hinder the user’s movements, which puts a restriction on the battery size and longevity.” Again, how do you compare your approach that needs to attach MI coils to the human body with other approaches that use embedded sensors on smart devices? In which one the proposed approach beats other technologies?*

RESPONSE: In the introduction, we have compared this approach with a Bluetooth IMU device in terms of cost. Also, the power consumption of conventional communication systems and systems working based on similar principles to the MI-system is reported. We have explained the characteristics of an MI system and its advantages. Please see the added material on pages 3-4, lines 85-123.

3. *Authors say: “inertial sensors such as accelerometers, gyroscope, and magnetometer, are attached to the user, and the collected data are transferred wirelessly to the processing node for motion detection.” Or, they say: “It eliminates the need for an extra wireless module and reduces power consumption by combining data collection and wireless signal transmission steps”. This is not clear to me. I couldn’t*

understand why accelerometers, gyroscope, and magnetometer, in my smartphone, or smartwatch, should send this data to a wireless node when they can process data on-device very well.

RESPONSE: Our statement here is related to a system that contains more than one device for activity recognition. As you have noted, one approach to capture motion is to use a single device placed on a fixed location of the user (e.g., smartwatch). Although studies have reported promising results on detecting human movements using, for example, smart watch's data, they still face challenges. Single devices cannot cover the whole body and fail to collect adequate information about the mobility of all body segments. Additionally, in systems relying on data from a single device, variations in position can have a significant effect on the performance or lead to the failure of the monitoring system. In other words, for applications where data from multiple body parts is required, single devices are not practical, and a network of sensors is needed. In the sensor networks, all nodes should send data to a central unit for processing. A more detailed explanation is provided in the introduction. Please see page 2, lines 22-44.

4. *Authors say: “the automatic recognition of complex human activities in daily life using multiple wearable sensors is still an open research topic”, and “Although sensor systems can overcome many disadvantages of the external approach, they still face many challenges”. But the references are some surveys from 2012 and 2013. There are many recent studies in this area that show interesting results in HAR using already embedded motion sensors in smart devices. The set of activities (walk, knock, lift, and throw) is a small set in size and not the most common and challenging activities. For example, I expected you evaluate the performance of some activities that are difficult to distinguish by other approaches, like walking in a flat area with walking downstairs or upstairs. Overall, the paper has many claims on the supremacy of the MI approach, but we don't see a fair comparison between the MI approach and current approaches like smart devices motion sensors. It is important to clearly know in what exact situation this MI approach beats other approaches. My suggestion is to compare with the refs. [24] or [28].*

RESPONSE: We included more types of activities to provide a more diverse set of actions. Machine learning-based classifiers are also applied to the data for performance comparison. The results are compared to recent studies reported on the same datasets as reference. It should also be noted that three activities used in the paper: knock, lift, and throw are very similar, and their results show the capability of the proposed system in distinguishing similar movements. These revisions have been included in the paper on pages 4-7, lines 271-308, Table 1-2, Figs. 4-5.

5. *Results reported in the Results section is better to be in a table or something more structured. Writing a paragraph full of numbers makes it hard to follow the main findings and compare the results.*

RESPONSE: Done. We have included more results for different classifiers and datasets, and the paper is revised accordingly. Please see pages 4-7, lines 271-308, Table 1-2, Figs. 4-5.

6. *As the techniques and methods for using MI coils to collect human motions data have already been presented in other authors' publications, like refs. [16], [17], [18], and [26], the main novelty of this paper seems to be applying RNNs for activity recognition based on data collected by MI coils. Thus, we expect to see more evaluation of this part of the paper. For example, what if we use a simpler and more lightweight ML model, like random forests or if we use a convolutional network than RNNs? We know*

RNNs give us better accuracy, but the cost of running RNNs should also be considered for real-time applications.

RESPONSE: In our previous works, we studied the MI system as a physical layer for wireless body area networks, and modeled the system as a two-port network. The accuracy of the model for different frequencies for multiple fixed locations and alignments was investigated, and matching networks were suggested for performance improvement. In those works, we did not propose the MI system for activity detection (MI-HAR); this concept is introduced in the present paper for the first time. Moreover, the model used in this paper is more accurate, and we have provided time-series measurements for model validation. Using the model, we synthesized the MI-motion data to investigate the capability of the MI system in detecting movements. This was not addressed in the previous papers. In summary, the whole proposed framework is the novelty of this work. We have now also included machine learning-based classifiers for comparison. The primary purpose of the classification stage is to show that the MI signal provides discriminative data for distinguishing activities from each other. We compared the classification methods, and results are shown in “Performance” and “Discussion” sections. Please see pages 4-7, lines 230-367.

Minor comments:

- 7. In the abstract and introduction, what is the exact role of “wireless sensor networks” in recognizing human physical activities? Why we need “wireless sensor networks” when we can simply use an application on the user's smartphone or their smartwatch?*

RESPONSE: We have explained in detail the advantages of sensor networks over a single node device for data collection. We referenced recent studies on this topic. Please see page 2, lines 22-44.

- 8. Moreover, in practice, where will these “Deep Recurrent Neural Networks” be run? It is very good to have a top-level architecture that explains these details.*

RESPONSE: We have mentioned in the paper that depending on the application and available processing units, the model should be modified. The network design is a trade-off between accuracy and computational complexity. We envision that one of the MI-HAR nodes can be used to process the data on-board, and that with proper optimization of the RNN (or other ML) architecture, the processing would be done in near-real-time with modest resources. We appreciate the reviewer's bringing up this important point, which is a practical matter that needs to be considered in building a realistic prototype of this technology.

- 9. It's better to use “average classifications accuracy” than “Success rate”, as it is the common and well-known term in ML literature.*

RESPONSE: Thank you for your suggestion. We changed the cross-validation methods to leave-one-subject out and updated the paper accordingly.

- 10. Authors say: “Required memory, security, and privacy of the collected personal data should be guaranteed by the system”. I couldn't find any evaluation or discussion to compare the security or*

privacy concerns of the proposed method with other approaches, like sensor-based one. Do we have some benefits in using MI coils in these specific concerns? It is good to have a brief discussion on these challenges.

RESPONSE: The principal characteristics of the MI system provide these advantages, which are now explained in more detail in the introduction. Please see pages 2-3, lines 77-123.

11. *“As long as the MI system operates in the induction regime, it can work well in any non-magnetic environment such as around the human body.” Please more explain this. Does this mean when the user is using some magnets or something that has some magnets into it, then the system will lose accuracy?*

RESPONSE: The MI communication is immune to permanent magnetic fields because it does not produce a time-varying magnetic flux, and therefore, an MI coil cannot detect it. Similar to the conventional systems in which electromagnetic pulses might affect their performance, these systems can also be affected, which does not disqualify the system from the intended routine operation. Furthermore studies have investigated interference in systems based on similar principles to mitigate interference, which can be used in this method as well (Explained in more detail in pages 3-4, lines 218-229). It worth mentioning that the MI-receiver receives different signal patterns during distinct movements, which is used to identify the actions. If a ferromagnet or any other interference changes the signals, all received MI signals would be affected, but they still will be distinctive.

REVIEWERS' COMMENTS:

Reviewer #1 (Remarks to the Author):

The paper addresses a very important and interesting problem of human activity recognition having a lot of useful applications. The authors correctly state that HAR is still an open scientific problem. The paper introduces a very interesting and novel idea of using a wireless system based on magnetic induction (MI). The signals obtained using this system are then processed by modern and sophisticated machine learning algorithms, i.e., deep recurrent neural networks. The results obtained on synthetically created dataset for four categories of physical activities, namely walk, knock, lift, and throw are convincing.

The idea of using magnetic induction for the acquisition of signals to be processed is very impressive, novel and definitely worth further scientific investigations. Following the hint from my review of the first version of the paper, the authors have significantly extended the experimental setup for validate the proposed approach. This led to much more sophisticated evidence-based conclusions. Summarising, the authors have proposed a very novel algorithm for human activity recognition that proved its robustness in a comprehensive evaluation phase.

Reviewer #2 (Remarks to the Author):

I would like to commend the authors for taking the time to prepare a detailed rebuttal to the reviewers' comments. I would also like to commend them for making substantial modifications to the manuscript in order to address the reviewers' comments. The quality of the material was substantially improved, and I would like to recommend that the manuscript be published in Nature Communications.

Reviewer #3 (Remarks to the Author):

The revised version of the paper addresses major issues raised by the reviewer.

Response to Referees

Reviewer #1

The paper addresses a very important and interesting problem of human activity recognition having a lot of useful applications. The authors correctly state that HAR is still an open scientific problem. The paper introduces a very interesting and novel idea of using a wireless system based on magnetic induction (MI). The signals obtained using this system are then processed by modern and sophisticated machine learning algorithms, i.e., deep recurrent neural networks. The results obtained on synthetically created dataset for four categories of physical activities, namely walk, knock, lift, and throw are convincing.

The idea of using magnetic induction for the acquisition of signals to be processed is very impressive, novel and definitely worth further scientific investigations. Following the hint from my review of the first version of the paper, the authors have significantly extended the experimental setup for validate the proposed approach. This led to much more sophisticated evidence-based conclusions. Summarising, the authors have proposed a very novel algorithm for human activity recognition that proved its robustness in a comprehensive evaluation phase.

RESPONSE: Thank you for your thoughtful review of our work and kind words.

Reviewer #2

I would like to commend the authors for taking the time to prepare a detailed rebuttal to the reviewers' comments. I would also like to commend them for making substantial modifications to the manuscript in order to address the reviewers' comments. The quality of the material was substantially improved, and I would like to recommend that the manuscript be published in Nature Communications.

RESPONSE: Thank you for your positive feedback and for recommending our paper for publication in Nature Communications.

Reviewer #3

The revised version of the paper addresses major issues raised by the reviewer.

RESPONSE: Thank you for your feedback.